# Heterologous expression and characterization of synthetic polyester-degrading cutinases from *Fusarium* spp. in *Aspergillus niger*

Jo-Anne Verschoor,[1] Mark Arentshorst,[1] Antonia J. G. Regensburg-Tuink,[1] Sjoerd J. Seekles,[2] Cees van den Hondel,[1] Johannes H. de Winde,[1] Arthur F. J. Ram[1]

**ABSTRACT** Cutinases are versatile enzymes with substrate promiscuity, making them promising candidates for the degradation of both natural and synthetic polyesters. While bacterial cutinases have been extensively studied, fungal cutinases remain underexplored, particularly in their enzymatic activity beyond their role in plant virulence. In this study, we investigated four cutinases from two *Fusarium* strains. Both strains displayed activity on Impranil-DLN, revealing their polyester-degrading potential. The strains were identified as *Fusarium oxysporum* (strain 38) and *Fusarium redolens* (strain 62). We characterized two cutinases per strain, tentatively called Cut1 and Cut5. Phylogenetic analysis revealed that both Cut1 clustered together in one branch where the Cut5 variants are closely related to the previously characterized bis(2-hydroxyethyl) terephthalate (BHET)-hydrolyzing enzyme *Fo*Cut5, providing structural insights and insights into their catalytic potential. We successfully expressed the Cut5 cutinases in *Aspergillus niger* using a CRISPR-Cas9-based multicopy integration system, resulting in enhanced degradation of Impranil-DLN and tributyrin. Using the same multicopy integration approach, transformants containing multicopy Cut1 variants were obtained but found to produce considerably lower amounts of Cut1, resulting in less activity and disabling further purification. The optimal substrate length, temperature, and pH for both Cut5 enzymes were determined. Additionally, we show activity of the purified Cut5 enzymes on synthetic substrates Impranil-DLN and BHET, suggesting that these fungal cutinases may be valuable for bioremediation. Accelerating the discovery of fungal cutinases and optimizing their expression systems holds promise for future strategies for polymer degradation to reduce agricultural and plastic waste.

**IMPORTANCE** Cutinases are promising enzymes for a spectrum of applications due to their substrate promiscuity toward both natural and synthetic polymers. This makes them strong candidates for the development of sustainable solutions to battle environmental pollution. Therefore, the successful production and characterization of novel cutinases is fundamental for understanding their mechanisms and potential applications. In this study, we have identified, produced, and characterized two cutinases from different *Fusarium* species using a multicopy integration system in *Aspergillus niger*. Structural characteristics and *in vivo* and *in vitro* enzyme activity provide a unique insight into the polyester-degrading activity of these enzymes and how they can contribute to more sustainable solutions to our current waste management and pollution challenges.

**KEYWORDS** cutinase, Impranil-DLN, *Fusarium*, *Aspergillus niger*

Plants utilize a combination of complex polyesters to form an external barrier known as the plant cuticle, which forms a hydrophobic layer that protects the plant

Address correspondence to Arthur F. J. Ram, a.f.j.ram@biology.leidenuniv.nl, or Johannes H. de Winde, j.h.de.winde@biology.leidenuniv.nl.

The authors declare no conflict of interest.

from desiccation, predators, and pathogens (1–3). This plant cuticle consists of a cutin matrix supplemented with other complex natural compounds like waxes and carbohydrates. Cutin is a highly heterogeneous polymer with a species-dependent structure (4). Typically, the basic polymer consists of glycerol molecules esterified to hydroxy- or epoxy fatty acids with C16 or C18 tails (3, 5). The abundance of specific monomers and crosslinks within the molecule results in a stable, dense, semi-permeable structure (5). While cutin is highly recalcitrant and resistant to various biotic and abiotic factors (1), plant-affiliated microbes have evolved specialized enzymes, known as cutinases, to depolymerize this protective layer.

Cutinases (EC 3.1.1.74) occur in a wide range of bacterial and fungal species and play a crucial role in plant infiltration, infection, and degradation. This group of enzymes belongs to the α/β-hydrolase family and hydrolyzes ester bonds within the cutin polymer (6, 7). The catalytic properties resemble those of esterases and lipases but are highly unique for their activity disregarding the presence of water-oil interfaces (8). The catalytic triad is highly conserved, invariably consisting of Ser, Asp, and His residues. The overall structure of cutinases has been well characterized through X-ray crystallography and NMR (8, 9). While the core structure is conserved, displaying characteristic sequence motifs, cutinases exhibit significant differences in optimal temperature, pH, activity, and turnover rates (7).

Fungi represent a particularly interesting source of cutinases due to their abundance across various ecological niches and variety of lifestyles. Many hemi-biotrophic fungi involved in plant infection secrete multiple cutinases, which play crucial roles within the infection cascade (7). *Fusarium* species, in particular, are known to express several cutinases with distinct roles during infection. In *Fusarium solani*, three highly homologous cutinases, Cut1, Cut2, and Cut3, differ mainly in regulation and topical expression levels. Cut2 is likely constitutively expressed under starvation conditions. The release of cutin monomers through Cut2 action subsequently strongly induces Cut1 and moderately induces Cut3 (10). Understanding these regulatory mechanisms is essential for optimizing screening conditions for the identification of cutinases.

Beyond their role in plant pathogenicity, cutinases exhibit unique substrate catalytic promiscuity, making them valuable enzymes across multiple research fields and industrial settings. They have been shown to be valuable in the textile and laundry industry, transesterification of fats and oils, and the production of phenolic compounds (8). Notably, several cutinases display promiscuous activity on synthetic polyesters like polybutylene adipate terephthalate (PBAT), polyethylene terephthalate (PET), polybutylene succinate, and the aliphatic polyester-polyurethane dispersion Impranil-DLN. Impranil-DLN is commonly used for textile coatings, thereby providing waterproofing, flexibility, and durability (11, 12). Despite the structural differences between synthetic and natural polymers, their shared hydrophobic nature and ester bonds have been shown to provide enough similarity to enable cutinases to act on these vastly different substrates (6). The plastics-active enzymes database (PAZY database) indicates that the most synthetic polyester-degrading cutinases were isolated from bacteria (13). Important bacterial polyesterases are the leaf compost cutinase (LCC) from an unidentified prokaryotic organism, *Thermobifida fusca* cutinase *Tf*Cut2, and the *Is*PETase from *Ideonella sakaiensis*. Each enzyme has been characterized extensively and optimized to increase activity significantly (14–17). In addition, several fungal cutinases have been described to have similar polyester-degrading activities (13, 18). Examples of cutinases isolated from filamentous fungi are the *Aspergillus niger* Cut3 (19), which is able to modify the surface of PET and polycaprolactone (PCL), and the cutinase from *Humicola insolens* with activity on PET and PBAT (20–23). Structural and functional studies of *Fusarium oxysporum* cutinase *Fo*Cut5 showed PET and PCL depolymerizing activity (24). *In silico* docking experiments further expanded its potential to degrade at least seven additional synthetic polymers (25). These findings underscore the significant potential of fungal cutinases for synthetic polymer degradation.

In this study, we identified and expressed four cutinases from two endophytic *Fusarium* strains using *A. niger* as a CRISPR-Cas9-based production platform (26). This system is designed to integrate multiple gene copies into the genome of *A. niger,* an established host for recombinant protein production, to obtain high protein yields. Additionally, several extracellular proteases have been deleted to avoid extracellular degradation of the protein of interest (26). We investigated the phylogeny, predicted structure, optimal conditions, and substrate preference. Understanding the substrate promiscuity of fungal cutinases could lay the foundation for the development of sustainable degradation and recycling of natural and synthetic polyesters.

## RESULTS

### Screening of endophytic *Fusarium* strains to identify cutinases

Plant-associated fungal strains might offer an interesting source of cutinases, often only studied in relation to plant pathogenicity. To investigate the polyester degradation capacity of plant-related fungal species, a collection of fungal strains isolated from plants was screened for growth on plates containing Impranil-DLN or bis(2-hydroxyethyl) terephthalate (BHET), a monomer of PET, to investigate possible polyester-polyurethane or PET-hydrolyzing activity. Strains displaying high Impranil degradation activity were selected and further analyzed. On Impranil-DLN, *Fusarium oxysporum* f. sp. *radicis-lyco-persici* (27, 28) showed a modest halo around the core of the colony. Importantly, the other *Fusarium* strains showed a wider clearance halo coinciding with the width of the mycelium (see Fig. 1a for an arrayed representation of all relevant growth plates). In addition, plates containing tributyrin and phenol red were inoculated, allowing the visualization of the degradation of lipids, which causes a lower pH, turning the phenol red to yellow. No lipase or BHETase activity was observed in any strain (Fig. 1a). *Fusarium* strains 38 and 62 were selected for further research, and ITS sequencing revealed 100% identity to *Fusarium oxysporum* for strain 38 and high similarity to *Fusarium fujikuroi* for strain 62 (100% coverage, 98.68% identity). Scavenging the *F. oxysporum* and *F. fujikuroi* genomes revealed two possible cutinases for each strain, tentatively called Cut1 and Cut5. The genes encoding both cutinases of strains 38 and 62 were amplified to obtain the complete sequence of each cutinase. Full genome sequencing of the #38 and #62 strains identified them as *Fusarium oxysporum* and *Fusarium redonens,* respectively. The detailed analysis of these genomes falls outside the scope of this paper. Sequence analyses revealed that the Cut1 protein from *F. oxysporum* and *F. redonens* varied at 21 amino acid positions between the two strains. The Cut5 enzymes were highly similar, differing by only three amino acids (Fig. 1b). Phylogenetic analysis showed that *Fusarium* cutinases cluster together and are distinctly different from the *Aspergillus* cutinases (Fig. 1c). Interestingly, the Cut1 proteins from *F. oxysporum* #38 (*Fo*38Cut1) and *F. redonens* #62 (*Fr*62Cut1) cluster in one branch as a separate group. The Cut5 enzymes share a branch with *Fo*Cut5, suggesting high similarity. An additional alignment to *Fo*Cut5 showed a percent identity of 99.57% for 38Cut5 and 98.26% for 62Cut5; therefore, these enzymes will hereafter be referred to as *Fo*38Cut5 and *Fr*62Cut5 (Data S1). Additionally, the signal peptide prediction, amino acid sequence, and structure comparison with *Fo*Cut5 uncovered that the first 16 amino acids most likely code for a signal peptide followed by a pro-peptide, which is only observed in the amino acid sequence and absent in the crystal structure (Fig. 1b) (24, 29). The pro-peptide of 38Cut1 is longer than the other pro-peptides and contains a repeating sequence. According to the approach of Dimarogona and colleagues, the signal peptide and pro-peptide were excluded to obtain the predicted structure and show the core similarities between the four cutinases and *Fo*Cut5 (Fig. 1d).

### Expression of *Fusarium* cutinases in *A. niger*

Using a recently developed CRISPR-Cas9-based multicopy integration system, three or four copies of each His-tagged versions of cutinase encoding genes (*Fo38cut1*, *Fr62cut1*,

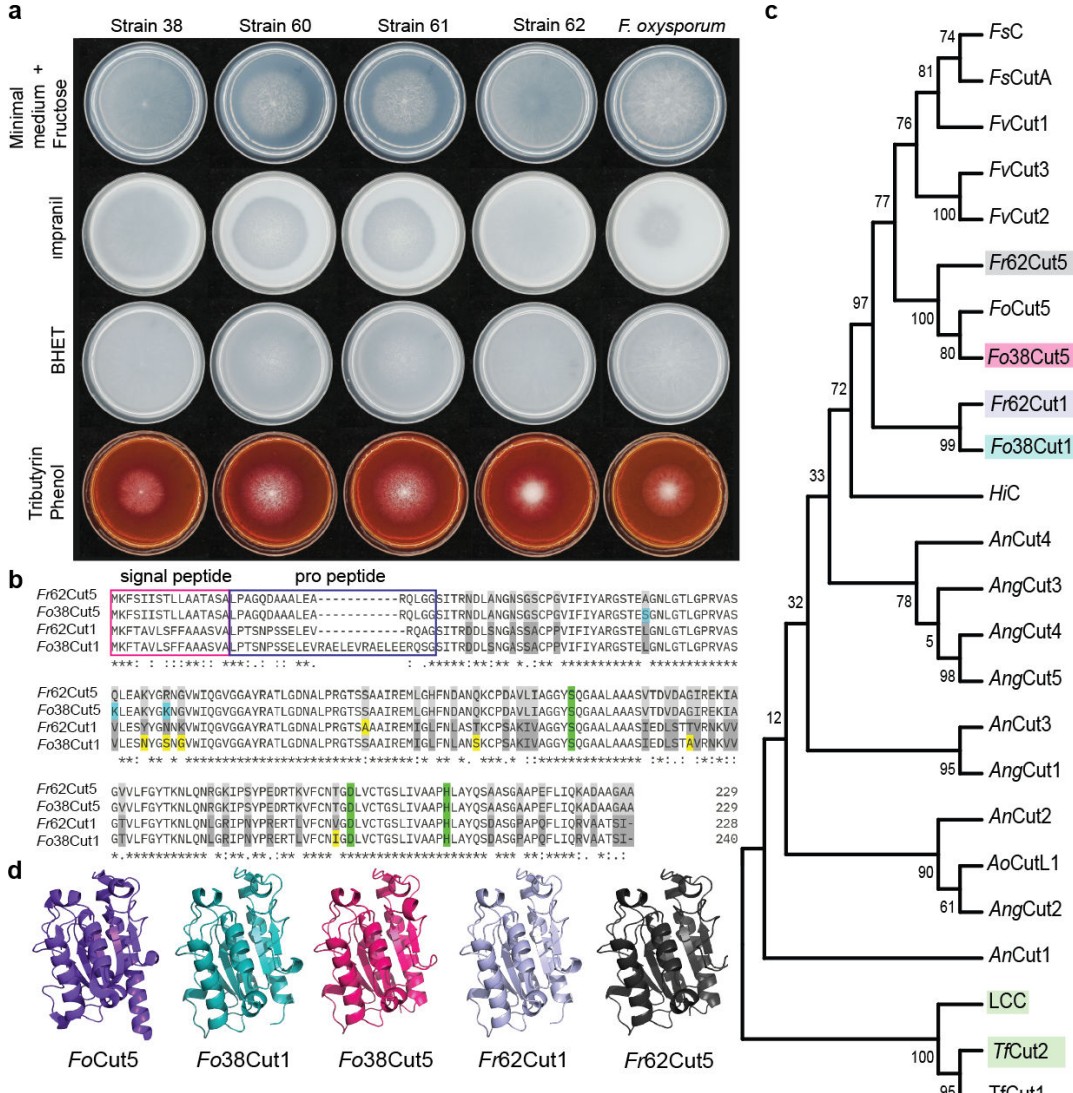

**FIG 1** Screening of endophytic *Fusarium* strains to identify novel cutinases. (a) Screening *Fusarium* strains for Impranil-DLN degradation, PETase activity (BHET), and lipase activity (tributyrin + phenol red). (b) Sequence comparison of *Fr*62Cut1, *Fr*62Cut5, *Fo*38Cut1, and *Fo*38Cut5. Amino acid differences among the different cutinases are highlighted in gray; when only one sequence differs, the amino acid in the Cut5 sequences was highlighted in turquoise, and the sequence differences in the Cut1 sequences were highlighted in yellow. The catalytic triad was highlighted in green. The predicted signal peptide was indicated using a magenta frame, and the pro-peptide was indicated using a dark blue frame. (c) Phylogenetic analysis of published fungal cutinases and novel *Fusarium* cutinases. The names and sequences are provided in Table 1 and Table S1. The novel cutinases are highlighted in gray (*Fr*62Cut5), magenta (*Fo*38Cut5), turquoise (*Fo*38Cut1), and lilac (*Fr*62Cut1). The control cutinases in the outgroup consisting of bacterial cutinases LCC, *Tf*Cut2 were highlighted in green. (d) The crystal structure of *Fo*Cut5 (violet) compared to the predicted AlphaFold models of *Fusarium* cutinases displayed in turquoise (*Fo*38Cut1), magenta (*Fo*38Cut5), lilac (*Fr*62Cut1), and gray (*Fr*62Cut5).

*Fo38cut5,* and *Fr62cut5*) were integrated in the *A. niger* genome at specific landing sites, which allow expression of the gene of interest from the strong glucoamylase promoter (26). Successful integration and number of integrations for each transformant was verified via PCR (data not shown). To analyze whether the introduction of multiple copies of the various cutinase encoding genes resulted in the production of active cutinases, transformants expressing the different cutinase genes were grown on minimal medium plates supplemented with 0.5% Impranil-DLN, 10 mM BHET, or 1% tributyrin with 0.01% phenol red (see Fig. 2a for an arrayed representation of all relevant growth plates). Degradation of Impranil-DLN provides insights into the degradation of this specific synthetic polymer, whereas BHET is a precursor for the degradation of PET. Tributyrin

**TABLE 1** Enzymes used for phylogenetic analysis

| Name in tree phylogenetic | Species | Accession number UniProt | Reference |
|---|---|---|---|
| AnCut1 | Aspergillus nidulans | Q5B2C1 | (30) |
| AnCut2 | Aspergillus nidulans | Q5AVY9 | (30, 31) |
| AnCut3 | Aspergillus nidulans | Q5AX00 | (30) |
| AnCut4 | Aspergillus nidulans | C8VJF5 | (30) |
| AngCut1 | Aspergillus niger | Supplemental data | (19) |
| AngCut2 | Aspergillus niger | Supplemental data | (19) |
| AngCut3 | Aspergillus niger | Supplemental data | (19) |
| AngCut4 | Aspergillus niger | Supplemental data | (19) |
| AngCut5 | Aspergillus niger | Supplemental data | (19) |
| AoAmi | Aspergillus oryzae | Q12559 | (32) |
| AoCutL1 | Aspergillus oryzae | P52956 | (33, 34) |
| FoCut5 | Fusarium oxysporum | X0BTD8 | (24) |
| FsC | Fusarium solani pisi | AAA33335.1 | (35) |
| FsCutA | Fusarium solani | Q99174 | (36) |
| FvCut1 | Fusarium vanetteniii | P00590 | (37) |
| FvCut2 | Fusarium vanetteniii | Q96UT0 | (10) |
| FvCut3 | Fusarium vanetteniii | Q96US9 | (10) |
| HiC | Humicola insolens | A0A075B5G4 | (20, 22, 23) |
| LCC | Compost | G9BY57 | (15) |
| TfCut1 | Thermobifida fusca | E5BBQ2 | (38) |
| TfCut2 | Thermobifida fusca | E5BBQ3_THEFU | (16, 38) |

confirms general lipase/cutinase activity and verifies successful enzyme expression. The parental *A. niger* strain (MA1048.1) did not display halos of degradation, indicating that it has no native activity. Integration of either *Fo38cut1* or *Fr38cut1* shows a modest halo on Impranil-DLN and an acidifying halo on tributyrin and phenol red, indicating lipase-like activity. The *cut5*-expressing transformants displayed a larger clearance halo on Impranil-DLN and a more intense yellow halo on the plates containing tributyrin and phenol red (see Fig. 2a). Integration of *Fo38cut5* and *Fr62cut5* also resulted in a slight morphological change and reduced sporulation phenotype on minimal medium plates with fructose, indicating some negative effect on growth and/or differentiation upon Cut5 production.

To quantify the amount of cutinase produced in the transformants, the strains were grown in liquid cultures and medium samples were analyzed using SDS-PAGE and Western blot. The analysis indicated that all four cutinases were produced. The molecular weight of the cutinase was as expected (around 25 kDa). The Western blot indicated that the proteins contain their C-terminal His-Tag. At the later time points (48 and 72 h after spore inoculation), some degradation of the Cut5 proteins was visible, which was probably induced by starvation-induced proteases. The analysis also revealed a much higher production of both *Fo*38Cut5 and *Fr*62Cut5 proteins compared to *Fo*38Cut1 and *Fr*62Cut1 (Fig. 2b). Noteworthy are the high levels of the cutinases in the spent medium after 48 h and 72 h of culturing (Fig. 2b and c). Since in some experiments degradation of cutinases was observed at the 72 h time point, it was decided to harvest spent medium at 48 h to avoid degradation of the enzymes. Enzyme assays using 48 h spent medium samples show significant Impranil-DLN degrading activity directly after inoculation with the samples containing Cut5 but no significant activity in the samples containing Cut1 ($n$ = 3, $P$ = 0.05, Data S2) (Fig. 2d).

## Purification and characterization of *Fo*38Cut5 and *Fr*62Cut5

To avoid interference of the background proteins, the heterologously expressed enzymes were purified via His-purification. The low amount of Cut1 present in the spent medium did not result in enough protein for further analysis. In contrast, Cut5 was successfully

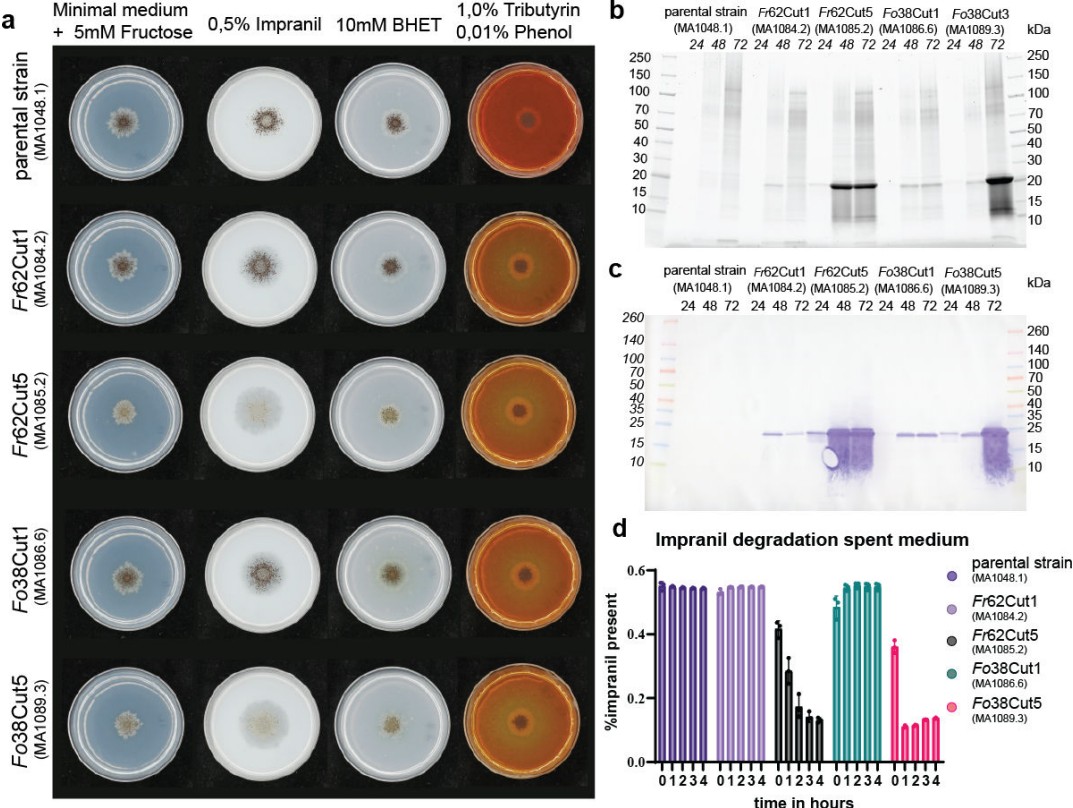

**FIG 2** Expression of *Fusarium* cutinases in *A. niger*. (a) Screening *A. niger* cutinase expression strains for Impranil-DLN degradation, PETase activity (BHET), and lipase activity (tributyrin + phenol red). (b) SDS-PAGE of concentrated (4×) spent medium of cutinase expression cultures at 24 h, 48 h, and 72 h. (c) Western blot using His-antibody of concentrated (4×) spent medium of cutinase expression cultures at 24 h, 48 h, and 72 h to confirm presence of cutinases. (d) Activity of the spent medium (1.5 ng/μL protein) on 0.5% Impranil-DLN. The medium control was indicated in violet. The spent medium samples were indicated by the enzyme present in lilac (*Fr*62Cut1), gray (*Fr*62Cut5), turquoise (*Fo*38Cut1), and magenta (*Fo*38Cut5). All error bars display the standard deviation, $n = 3$, significance determined using one-way analysis of variance. Significance ($P = 0.05$) can be found in Data S2.

purified and further analyzed (Fig. 3a). Preferred substrate length of both purified enzymes was determined using para-nitrophenyl substrates with different tail lengths (C4 until C16). Esterase activity was observed on all substrate lengths for both Cut5; for *Fo*38Cut5, nitrophenyl decanoate (C12) showed the highest activity (Fig. 3b). Para-nitrophenyl dodecanoate (C12) was chosen for the identification of the optimal conditions according to the methods of Altammar and colleagues (19). For both *Fo*38Cut5 and *Fr*62Cut5, the optimal pH was pH 7, and the optimal temperature was room temperature (20°C) (Fig. 3c, Data S3). These findings are consistent with the optimal growth conditions for *Fusarium* (39).

Impranil-DLN degradation assays were performed using the optimal conditions. Within 4 h, the majority of the Impranil-DLN was degraded by either Cut5 proteins. Both cutinases show a similar speed of Impranil-DLN degradation as the LCC and degrade Impranil-DLN faster than positive control *Tf*Cut2 (under suboptimal conditions). Further optimization of Impranil-DLN degradation confirmed that the determined optimal conditions were consistent with the pNp degradation conditions (Data S4). Additionally, significant BHET degradation was observed for *Fo*38Cut5 after 24 h. For *Fr*62Cut5, no significant BHET degradation was observed. Since the assay relies on a pH shift triggered by acid release during hydrolysis, it is possible that reduced enzyme stability resulted in insufficient BHET hydrolysis to produce a detectable signal (Fig. 3d; Data S2).

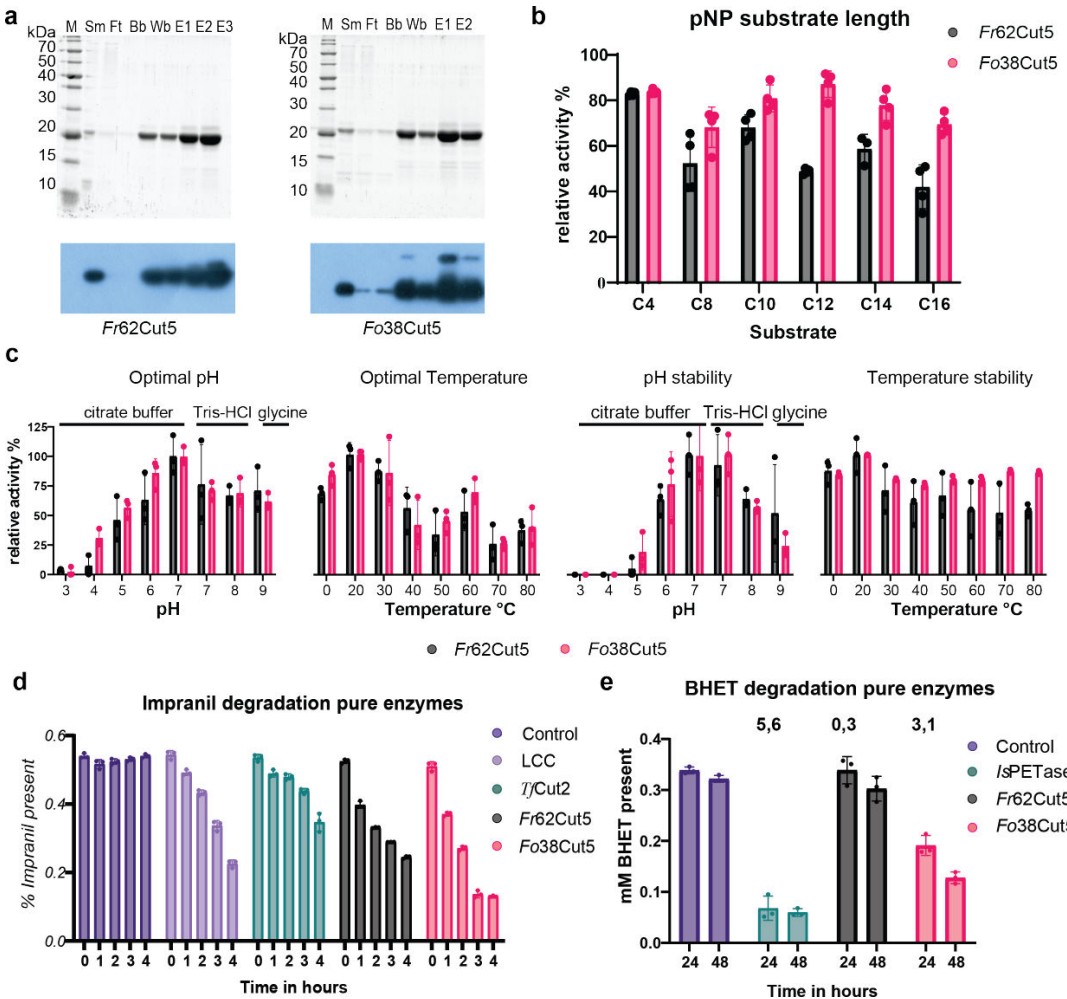

**FIG 3** Purification and characterization of *Fr*62Cut5 and *Fo*38Cut5. (a) SDS-PAGE (top) and Western blot (bottom) of purification of 62Cut3 (left) and 38Cut3 (right) spent medium samples (Sm), flowthrough (Ft), binding buffer (BB) wash buffer (Ws), and elution 1–3 (E1–3) were checked for the presence of the protein. (b) Substrate length determined using para-nitrophenyl substrates with different tail lengths (C4, C8, C10, C12, C14, and C16) corrected for maximum color conversion; *Fr*62Cut5 was displayed in dark gray, and *Fo*38Cut5 is displayed in magenta. (c) Optimal conditions determined using para-nitrophenyl dodecanoate; relative activities are displayed. The absolute values were similar for both Cut5 and are displayed in Data S3. (d) Activity of the pure enzyme samples (1 ng/µL protein) on 0.5% Impranil-DLN. The buffer control was displayed in violet, the control enzyme LCC in lilac, *Tf*Cut2 in turquoise, *Fr*62Cut5 in gray, and *Fo*38Cut5 in magenta. All error bars display the standard deviation, *n* = 3. Significance was determined using one-way analysis of variance (ANOVA); significance can be found in Data S2. (e) Enzymatic BHET degradation using colorimetric assay after 24 and 48 h. The buffer control is displayed in violet, the control enzyme *Is*PETase in turquoise, *Fr*62Cut5 in gray, and *Fo*38Cut5 in magenta. All error bars display the standard deviation, n = 3. Significance was determined using one-way ANOVA ($P = 0.05$). Significance can be found in Data S2. The average turnover rates (µmol/h/mg enzyme) are displayed above the bars graphs. The exact values are displayed in Table S2.

## DISCUSSION

Cutinases exhibit unique substrate promiscuity, making them promising enzymes for the depolymerization of both natural substrates and synthetic polymers. While numerous bacterial cutinases have been identified and characterized (13–16, 40, 41), fungal cutinases have primarily been studied in the context of virulence, with only a handful investigated for their enzymatic activity (13, 20, 35).

In this study, we have selected two plant-associated *Fusarium* strains with activity on polyester Impranil-DLN and identified them via ITS sequencing and full genome sequencing. From each identified strain (*F. oxysporum* and *F. redolens*), two cutinases named Cut1 and Cut5 were analyzed in more detail. Phylogenetic analysis showed both

Cut1 proteins form their own branch in the phylogenetic tree, whereas the Cut5 proteins are closely related to well-studied BHET-hydrolyzing cutinase FoCut5, providing us with a crystal structure (24). The crystal structure of FoCut5 combined with a predicted signal sequence (29) revealed the presence of a pro-peptide, which is absent in the mature protein but crucial for its expression, secretion, or production (24, 42, 43). By excluding the signal peptide and pro-peptide, the mature protein structures were predicted using AlphaFold (44), revealing four structures highly similar to the crystal structure of FoCut5 hinting toward BHETase activity.

Following gene identification, the cutinases were expressed in *A. niger* using an efficient multicopy integration system (26). The expression of these cutinases enables the ability of the transformed *A. niger* strain to degrade Impranil-DLN and tributyrin. Surprisingly, no activity was detected on the PET model substrate BHET, despite the expectations that both Cut5 proteins would exhibit activity due to their high sequence identity with FoCut5. SDS-PAGE and Western blot analysis confirmed the production of all four cutinases, with peak concentrations observed at 48 h. However, after 72 h of cultivation, protein levels in the spent medium often declined, and results varied between cultivation rounds. The Cut5 variants were highly produced in *A. niger*, whereas Cut1 variants exhibited significantly lower protein yields despite being under the regulation of the strong glucoamylase promoter. We hypothesize that this may be due to the highly divergent pro-peptides. Since pro-peptides are crucial for successful expression, incorrect recognition of these sequences in a heterologous system could negatively impact production (42, 43, 45, 46). Future studies could explore replacing the signal and pro-peptide sequences with endogenous ones to assess their impact on protein expression and interspecies compatibility.

Previous research into the production of heterologous cutinases in *Saccharomyces cerevisiae* uncovered several other causes of impaired expression, secretion, and/or production. For example, hydrophobic regions within cutinases can be misfolded, causing them to become trapped in the endoplasmic reticulum (ER). In the ER, the misfolded proteins are degraded via proteasomal degradation, thereby avoiding protein aggregation (47). Follow-up work indicated that the interaction between hydrophobic cutinases and immunoglobulin heavy-chain binding protein (BiP) might be the reason for retention in the cell (47). Other work revealed that glycosylation can be used to improve protein secretion and production. By introducing additional *N*-glycosylation sites in the protein, the production of the *Fusarium* cutinase was improved in *S. cerevisiae* (48). Additional experiments are needed to understand the low secretion levels of the Cut1 proteins in *A. niger*. Analysis of mycelium samples of strains expressing Fo38Cut1 or Fr62Cut1 showed faint bands after 24 h on Western blot, indicating the presence of Cut1 either intracellularly or attached to the cell wall. After 48 h, no bands were being observed, suggesting no accumulation of the Cut1 in the mycelium and possible degradation of the protein (Fig. S1). Efficient ER-associated degradation of misfolded cutinase or mistargeting cutinase to the vacuole and subsequent degradation are likely reasons why the production of both Cut1 proteins is low. Further post-translational and proteomic analysis of the intracellular cellular content would be required to understand why Cut1 showed a much lower protein level. Unfortunately, the low yield of Cut1 prevented further purification and characterization.

We further investigated the optimal conditions and activity of both pure Cut5 enzymes on Impranil-DLN and BHET. While both were active on Impranil-DLN, only Fo38Cut5 displayed activity on BHET. This difference may correlate with a minor sequence variation causing difference in substrate binding between Fr62Cut5 and Fo38Cut5 (see Fig. 1b), where Fo38Cut5 is phylogenetically closer to FoCut5, which has been shown to exhibit activity on BHET. On the other hand, the activity difference may relate to variation in available enzyme concentrations in the assays, as well as to variation in enzyme stability, as Fr62Cut5 exhibited lower activity in all assays compared to Fo38Cut5. The two enzymes share similar optimal conditions, with peak activity at room temperature (20°C) and pH 7, aligning with the optimal growth conditions of *F.*

*oxysporum* (39). Given their similarity to *Fo*Cut5, it would be valuable to express the different enzymes in the same host and to test all three enzymes on additional synthetic substrates, as suggested by Vinicius and colleagues (25).

Since this was merely an *in silico* study, showing the potential degradation potential of *Fo*Cut5 on nine synthetic substrates, the actual activity still needs to be confirmed. If the enzymes indeed possess such a broad substrate range, further studies will be required to evaluate their robustness, scalability, and long-term stability under industrial and environmental conditions. This will reveal whether those pro-enzymes would be potential candidates for the development of bioremediation processes or enzymatic polymer recycling (49, 50). In addition, the moderate substrate promiscuity of these enzymes makes them interesting candidates for further optimization toward several different natural and synthetic polyesters. Using site-directed mutagenesis or random mutagenesis could lead to different enzyme variants optimal for different substrates or substrate mixes while maintaining the same optimal conditions (51). To further improve activity, binding domains could be attached to increase the proximity of the enzyme to the substrate, allowing for enhanced activity (52).

Apart from enzyme optimization, the CRISPR-Cas9-based multicopy integration system could also be a promising asset for the production of specialized enzyme cocktails. The *A. niger* multicopy expression system presents an efficient strategy to produce multiple enzymes in one strain and obtain a customized enzyme cocktail. The integration system consists of a total of 10 potential landing sites that can be filled in different rounds of transformation with different enzymes. By integrating various enzymes into different landing sites, it is possible to control and vary gene copies and thereby achieve controlled expression levels to optimize enzyme ratios for specific applications. Since activity can potentially be enhanced by synergy between enzymes as shown by Carniel and colleagues, combining enzymes might be an important approach (22). The combination of the *H. insolens* cutinase (HiC) and *Candida antarctica* lipase improved the PET degradation efficiency significantly. Developing enzyme cocktails could improve polymer degradation efficiency (22, 53). Since the strain has a relatively clean background, purification of these cocktails would be minimal, making the process more feasible. This could be a very promising method to design a variety of enzyme cocktails for the laundry, paper, textile, and food industry (54, 55).

Overall, we successfully identified, expressed, and produced four *Fusarium* cutinases in *A. niger* using a multicopy integration system. Two Cut5 variants were characterized for their optimal conditions, substrate preference, and moderate substrate promiscuity toward synthetic substrates. Fungal cutinases may offer a promising future approach for degrading agricultural and plastic waste, opening new avenues for bioremediation and sustainable polymer recycling.

## MATERIALS AND METHODS

Final concentrations are displayed between brackets behind the mentioned compound.

### Strains and growth conditions

All strains used in this study are listed in Table S3. *Fusarium* endophytes were isolated and made available by Prof. Dr. C.A.M.J.J. van den Hondel (Institute of Biology Leiden, Leiden University). The strains were isolated from the plants *Paris polyphylla* var. *yunnanensis* and *Dioscorea nipponica* Makino. *A. niger* strains were grown in liquid or solidified (containing 1.5% [wt/vol] Scharlau agar) minimal medium (MM) or in complete medium (CM) as described (56). Plate assays were performed by spotting 5 µL of $2 \times 10^5$ spores/mL on MM with 5 mM fructose. To examine Impranil-DLN degradation, BHETase and lipase activity, MM + 5 mM fructose was supplemented with 0.5% Impranil-DLN (11), 10 mM BHET (Sigma Aldrich, Cas: 959-26-2, PN 465151), or 1% tributyrin and 0.01% phenol red. *Fusarium* plates were incubated for 7 days at 21°C, and *A. niger* was grown at 30°C for 6 days. To produce cutinases, *A. niger* strains were cultured in 100 mL of

CM with $1 \times 10^6$ sp/mL at 30°C, and medium samples were taken after 24, 48, and 72 h. *Escherichia coli* DH5α was used for plasmid construction and cultured at 37°C in Luria-Bertani medium, with ampicillin (100 µg/mL).

## Gene identification

Genomic DNA of *Fusarium* strains 62 and 38 was isolated as described (56). ITS PCR was performed using primers V9g and ITS4 (see Table S4) and genomic DNA of strains 62 and 38 as the template. Degenerate primers (see Table S4) were used to PCR amplify cut1 and cut5 from *Fusarium* strain 62 and 38. PCR products were sequenced at Macrogen.

## Phylogenetic analysis

Sequences were subtracted from the PAZY.eu database, literature, and UniProt. The sequences are provided in Table S1. The protein names, accession codes, and references are provided in Table 1. The tree was constructed using MEGA11 in default settings using a bootstrap of 500 (57).

## Structure prediction and comparison

Protein structures were predicted using the amino acid sequences of the Cut1 and Cut5 enzymes, excluding predicted signal sequence default settings of AlphaFold2 (44) The structures were overlaid and aligned with the structure of *Fo*Cut5 to identify the pro-peptide (24).

## Expression constructs and cloning

Cutinase expression cassettes were constructed according to Arentshorst and colleagues (26). Cutinase genes were PCR amplified and fused to *A. niger* PglaA and TglaA fragments (see Table S4). Fusion PCR products were ligated into pJet1.2blunt (Thermo Fisher Scientific) and sequenced. Donor DNA for transformation was isolated by restriction of the expression cassettes with *Pme*I.

## Transformation of *A. niger* and analysis of transformants

*A. niger* transformants were obtained by selection for hygromycin resistance using a final concentration of 200 µg/mL hygromycin (56). Transformation of the recipient strain MA1048.1 was performed using both the pFC332-sgRNA-KORE2 and KORE3 plasmids (both 5 µg DNA) and PglaA-cut-6xHis-TglaA as donor DNA (3 µg DNA) (26). MA1048.1 is a derivative of strain MA1029.4, in which a mutation in the AmyR transcription factor is introduced to optimize expression from the PglaA promoter (M. Arentshorst and A.F.J. Ram, unpublished data). Primary transformants were purified, and plasmid loss was induced according to Arentshorst and colleagues (26). Correct integration of the donor DNA at four different loci was verified by isolating genomic DNA (56) and subsequent diagnostic PCRs (see Table S4). Strain MA1084.2 contains three copies of 62-cut1-HIS, strain MA1085.2 contains four copies of 62-cut5-HIS, strain MA1086.6 contains four copies of 38-cut1-HIS, and strain MA1089.3 contains three copies of 38-cut5-HIS (see Table S3).

## SDS-PAGE and Western blot

*A. niger* strains were cultured in 100 mL of CM with $1 \times 10^6$ sp/mL at 30°C, and medium samples were taken after 24, 48, and 72 h. Cultures were centrifuged at $4,000 \times g$ for 10 min, and the supernatant was filtered through a 0.2 µm filter. The supernatant was concentrated four times using a Viaspin column (Sartorius 10 kDas).

Overall, all SDS-PAGEs contained 12% acrylamide and were run for 20 min at 70 V to stack the proteins on the gels. Furthermore, the gel was run at 150 V until the loading dye reached the bottom of the gel. SDS-PAGE gels were stained with SYPRO Ruby (Thermo Fisher, S12000) or One-Step Lumitein UV Protein Gel Stain (VWR, 21004).

For the Western blot, the gels were transferred using a Bio-Rad Trans-Blot Turbo and the corresponding transfer packs (1704157EDU) according to the mixed gel protocol of Bio-Rad. The gel was washed using Tris-buffered saline (TBS) buffer and blocked using TBS with 0.5% Tween 20 (TBST) buffer containing 1% Elk milk. The blot was blocked for approximately 90 min. His-antibody was added to a final concentration of 1 µg/mL and incubated overnight (Thermo Fisher, K953-01). The blot was rinsed with water and washed four times with TBST. The blot was then incubated with luminol for 1 min, dried, and developed on X-ray film or treated with TMB Enhanced One Component horse radish peroxidase Membrane Substrate (Sigma, T9455).

## Enzyme purification

For purification of Cut5, *A. niger* strains MA1085.2 and MA1089.3 were cultured in 100 mL of CM with $1 \times 10^6$ sp/mL at 30°C, and the medium was harvested after 48 h. To obtain purified Cut3 variants, His-affinity purification using stationary column systems was performed, following standard His-tag purification protocols. During purification, the column was washed with 10 mL 20 mM Tris-HCl, 0.5 M NaCl containing 30 mM imidazole pH 8, and eluted with the same buffer containing 1 M imidazole and 0.3 M NaCl. After purification, the samples were washed with 50 mM Tris-HCl, 0.5 M NaCl using concentrator columns. The enzymes were stored in 25 mM Tris-HCl, 0.25 M NaCl, and 40% glycerol in −20°C.

## Enzyme assays

The concentration of enzyme was estimated using the Bradford method (Bio-Rad, Bradford 1× Dye Reagent 5000205). The assays regarding the substrate lengths were conducted using para-nitrophenyl substrates with different tail lengths provided by Sigma-Aldrich (Table S5).

To account for variations in color conversion across substrates, the maximum color release was determined by adding an excess of previously purified *Tf*Cut2. This maximum value was set to 100%, and the relative activity was calculated accordingly. The standard esterase/cutinase activity was tested using para-nitrophenyl substrates according to (Sigma-Aldrich, 61716) the protocol of Altammar and colleagues, with minor adjustments. For the determination of the optimal conditions, para-nitrophenyl dodecanoate was used as substrate. For optimal pH determination, 50 mM citrate buffers ranging from pH 3 to pH 7 were used, and for pH 7 and 8, 50 mM Tris-HCl buffer was used. pH 9 was achieved using a 50 mM glycine buffer. The incubation step of 10 min was prolonged to 1 h. The reaction was terminated using 0.1 M sodium carbonate (19).

The Impranil-DLN assay was carried out as follows. Ten microliters of 10% Impranil-DLN (0.5%) was combined with 50 mM citrate buffer, pH 7, and 10 µL of enzyme mix (20 ng/µL pure, 30 ng/µL spent medium) in a 96-well plate. The plate was measured every hour for 6 h.

The colorimetric assay BHET assay was performed according to the methods of Beech and colleagues (58). A total of 2 µg/mL of the enzyme was incubated with 0.5 mM of BHET for 24 and 48 h and measured at 615 nm in the Tecan M Spark. A reference line was made by adding BHET in concentrations from 0.5 mM to 0 mM in steps of 0.1 mM; an excess of *Tf*Cut2 was added to convert all BHET to mono-(2-hydroxyethyl) terephthalic acid (59). Micromoles per hour per milligram of enzyme were calculated using the absolute values of the triplicates. The amount of BHET degraded was calculated by deducting the absolute numbers from the average control value after 24 h, and then, this was converted back to micromoles per hour per milligram of enzyme.

## Data analysis

The amount of Impranil-DLN or BHET degraded was calculated with GraphPad using the above-mentioned references. The statistical analysis consists of a one-way analysis of variance using the default setting of GraphPad Prism ($P = 0.05$, $n = 3$). A comparison was

made between the means of the control and each enzyme treatment, providing insights into the significance of the Impranil-DLN or BHET-degrading activity of each enzyme. The outcomes of the statistical analysis are provided in Data S2. The error bars in the figures represent the standard deviation.

## ACKNOWLEDGMENTS

We would like to thank Mia Urem, Jeremy van Wijk, and Nubia Wassenaar who have contributed to the characterization of the *Fusarium* strains.

## AUTHOR AFFILIATIONS

[1]Department of Molecular Biotechnology, Institute of Biology, Leiden University, Leiden, the Netherlands

[2]Department of Molecular and Cellular Biology, University of Geneva, Geneva, Switzerland

## AUTHOR ORCIDs

Jo-Anne Verschoor http://orcid.org/0000-0003-1177-1831
Johannes H. de Winde http://orcid.org/0000-0002-7999-9317
Arthur F. J. Ram http://orcid.org/0000-0002-2487-8016

## AUTHOR CONTRIBUTIONS

Jo-Anne Verschoor, Conceptualization, Data curation, Investigation, Methodology, Validation, Visualization, Writing – original draft | Mark Arentshorst, Data curation, Investigation, Methodology, Visualization, Writing – original draft | Antonia J. G. Regensburg-Tuink, Data curation, Investigation, Methodology | Sjoerd J. Seekles, Data curation, Investigation, Validation | Cees van den Hondel, Conceptualization, Investigation, Methodology | Johannes H. de Winde, Conceptualization, Supervision, Writing – review and editing, Validation | Arthur F. J. Ram, Conceptualization, Project administration, Supervision, Writing – review and editing

## ADDITIONAL FILES

The following material is available online.

### Supplemental Material

**Supplemental material (Spectrum02177-25-s0001.pdf).** Data S1 to S4; Tables S1 to S5; Fig. S1.

### Open Peer Review

**PEER REVIEW HISTORY (review-history.pdf).** An accounting of the reviewer comments and feedback.

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
