## [Reviewer comments · Microbiology Spectrum]

Microbiology Spectrum

Heterologous expression and characterization of synthetic polyester-degrading cutinases from *Fusarium spp.* in *Aspergillus niger*

Jo-Anne Verschoor, Mark Arentshorst, Antonia Regensburg-Tuink, Sjoerd Seekles, Cees van den Hondel, Johannes de Winde, and Arthur Ram

Corresponding Author(s): Arthur Ram, Leiden University Press

Review Timeline:

Submission Date:	August 12, 2025
Editorial Decision:	September 12, 2025
Revision Received:	September 12, 2025
Accepted:	September 19, 2025

Editor: Jeffrey Gralnick

Reviewer(s): The reviewers have opted to remain anonymous.

Transaction Report:

DOI: <https://doi.org/10.1128/spectrum.02177-25>

Re: Spectrum02177-25 (**Heterologous expression and characterization of synthetic polyester-degrading cutinases from *Fusarium spp.* in *Aspergillus niger***)

Dear Prof. Arthur Ram:

Based on your responses and revisions to the prior round of review at AEM, I am pleased to inform you that your manuscript has been editorially accepted for publication. However, there are a few additional questions in the submission form that need to be answered before the final decision. Once these are completed, please return your submission so that I can move your paper forward to acceptance.

Revision Guidelines

Sincerely,
Jeffrey Gralnick
Senior Editor
Microbiology Spectrum

Re: Spectrum02177-25R1 (**Heterologous expression and characterization of synthetic polyester-degrading cutinases from *Fusarium spp.* in *Aspergillus niger***)

Dear Prof. Arthur Ram:

Your manuscript has been accepted, and I am forwarding it to the ASM production staff for publication. Your paper will first be checked to make sure all elements meet the technical requirements. ASM staff will contact you if anything needs to be revised before copyediting and production can begin. Otherwise, you will be notified when your proofs are ready to be viewed.

Sincerely,
Jeffrey Gralnick
Senior Editor
Microbiology Spectrum